# SABER: Continual Learning with Representation Conflict Management

Xuandi Luo [1]   Huaidong Zhang [1]   Yi Xie [1]   Shengfeng He [2]

## Abstract

Continual learning seeks to develop models capable of acquiring new tasks sequentially while retaining prior knowledge. A central challenge in this setting is managing inherent knowledge conflicts that arise as overlapping or contradictory information is introduced across tasks. While parameter-efficient fine-tuning (PEFT) techniques, particularly those based on Low-Rank Adaptation (LoRA), have shown promise by reducing interference through parameter isolation or modular architectures, they often treat conflict as something to avoid rather than address directly. In this work, we propose *S*ubspace-*A*ligned *B*alanc*e*d *R*ecomposition (SABER), a novel method that reframes continual learning as a problem of structured conflict management. SABER introduces a unified subspace alignment framework to support shared task representations, decomposes task-specific knowledge into orthogonal components to preserve distinct information, and recomposes them using an energy-aware balancing mechanism that coordinates contributions without compromising stability. Extensive experiments across multiple continual learning benchmarks show that SABER achieves performance on par with or surpassing state-of-the-art methods, offering a principled approach that directly addresses the root cause of forgetting by managing representational conflict.

## 1. Introduction

Continual learning enables models to acquire knowledge from sequential tasks over time, which is essential for real-world applications such as autonomous driving, personalized recommendation, and intelligent assistants (Aljundi et al., 2019; Li & Hoiem, 2017; Meng et al., 2025; Jiang

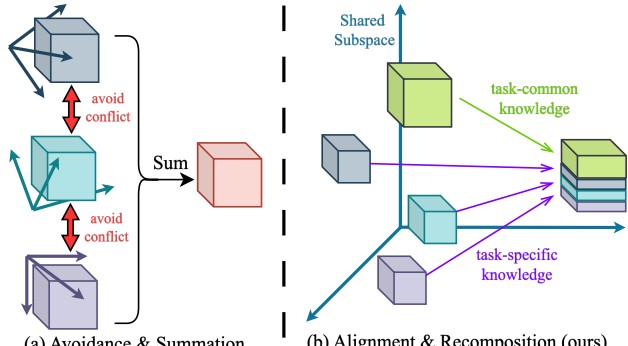

*Figure 1.* Conceptual comparison. (a) Conventional methods avoid interference via isolation, then simply sum parameters from different subspaces. (b) In contrast, our SABER proactively aligns tasks to a shared subspace for structured knowledge decomposition and balanced recomposition.

et al., 2026; Zhang et al., 2026). A central challenge in this setting is catastrophic forgetting (McCloskey & Cohen, 1989), where learning new tasks interferes with previously acquired knowledge. Effective continual learning therefore requires a delicate balance between stability (retaining past knowledge) and plasticity (adapting to new information) (Parisi et al., 2019). This problem is typically studied under various scenarios, including task-incremental learning, where task identities are available at test time, and class-incremental learning, where models must distinguish among all classes without explicit task labels (Wang et al., 2024; van de Weijer, 2021). The latter is particularly challenging and has received increasing attention due to its practical relevance.

With the advent of large-scale pre-trained models, continual learning has increasingly adopted parameter-efficient fine-tuning (PEFT) techniques, which adapt only a small set of parameters while keeping the base model frozen (Li & Liang, 2021; Jia et al., 2022). These methods, including prompt tuning (Lester et al., 2021), adapter modules (Houlsby et al., 2019), and Low-Rank Adaptation (LoRA) (Hu et al., 2022), offer computational efficiency and have shown strong performance in class-incremental settings (Khan et al., 2023; Wang et al., 2022b; Smith et al., 2023). By injecting trainable modules into frozen pre-trained backbones, PEFT methods mitigate forgetting while enabling rapid adaptation, positioning them as a promising direction

[1]South China University of Technology [2]Singapore Management University. Correspondence to: Huaidong Zhang <huaidongz@scut.edu.cn>.

*Proceedings of the 43rd International Conference on Machine Learning*, Seoul, South Korea. PMLR 306, 2026. Copyright 2026 by the author(s).

for scalable continual learning.

However, despite the empirical success of LoRA-based approaches, they face a core conceptual limitation in managing the tension between task-shared and task-specific representations. Existing methods often enforce orthogonality via soft regularization (Wang et al., 2023), gradient projection with memory overhead (Liang & Li, 2024), or orthonormal initialization (Wang et al., 2025), or resort on modular architectures like per-task LoRAs (He et al., 2025; Zhou et al., 2024), as shown in Fig. 1a. While such strategies reduce interference, they overlook a critical insight: representational conflicts across tasks are often inherent and cannot be fully avoided. As new tasks introduce competing structures or reuse shared capacity in conflicting ways, overlapping subspaces can result in destructive interference that simple isolation strategies fail to address. Enforcing artificial separation may inhibit positive transfer, while modular approaches often lack coordination mechanisms to integrate task-specific and shared knowledge effectively. This reveals a fundamental need to move beyond interference avoidance toward a more principled framework for handling inevitable representational conflicts.

To address this, we propose Subspace Alignment and Balanced Recomposition (SABER), a novel framework for continual learning that directly tackles the challenge of representation conflict. SABER is built on the core insight that conflict across tasks should not be eliminated but instead managed through structured coordination. As shown in Fig. 1b, our approach departs from conventional orthogonal or modular designs by introducing a unified subspace alignment mechanism that facilitates a shared representational space while preserving task-specific distinctions. By explicitly modeling and aligning representational subspaces across tasks, SABER promotes compatibility among conflicting components. It further introduces knowledge decomposition and recomposition to disentangle and reorganize different knowledge components in a task-aware manner. Finally, an energy-aware balancing module dynamically calibrates the contributions of shared and private components, ensuring stable and efficient learning across tasks. Together, these elements enable SABER to achieve improved knowledge integration and retention in sequential settings, offering a principled solution to the stability-plasticity dilemma in continual learning. Our key contributions are summarized as follows:

- We propose SABER, a continual learning framework that shifts the focus from avoiding interference to managing inherent representation conflicts.

- We develop a unified subspace alignment framework with orthogonal initialization that provides a shared representation space for cross-task knowledge integration.

- We introduce a principled decomposition and balanced recomposition mechanism that separates task-common and task-specific knowledge and recombines them for effective sequential learning.

- Through extensive experiments on multiple benchmarks, we show that SABER achieves state-of-the-art performance with strong parameter efficiency.

## 2. Related Work

**Parameter-efficient fine-tuning (PEFT)** adapts pre-trained models to downstream tasks by introducing a small set of trainable parameters, while keeping the original model weights fixed. This approach significantly reduces memory and computational costs compared to full fine-tuning, often without compromising performance. Early PEFT methods such as Adapters (Houlsby et al., 2019), insert lightweight modules into Transformer layers and train only these additions. Prompt-based methods, including prompt tuning (Lester et al., 2021) and prefix tuning (Li & Liang, 2021), learn a set of tunable tokens or embeddings prepended to the input sequence. LoRA (Hu et al., 2022) takes a different approach by injecting low-rank updates directly into the weight matrices of the attention layers, allowing efficient fine-tuning without altering the original weights. Although originally developed in NLP, PEFT techniques have been successfully adapted to computer vision. For instance, Visual Prompt Tuning (VPT) (Jia et al., 2022) and Adapter-Former (Chen et al., 2022) apply these ideas to Vision Transformers (ViTs), achieving competitive performance while training only a small fraction of the parameters. These methods form the foundation for more recent developments in continual learning with frozen pre-trained models.

**Continual learning** addresses the challenge of learning from a sequence of tasks without forgetting previous ones. Traditional approaches fall into three main categories. Expansion-based methods (Hung et al., 2019; Li et al., 2019; Wang et al., 2022a; Yan et al., 2021; Luo et al., 2025) dynamically grow the model's architecture to accommodate new tasks. Regularization-based methods (Aljundi et al., 2018; Jung et al., 2020; Kirkpatrick et al., 2017; Zenke et al., 2017) constrain updates to important parameters via additional loss terms. Memory-based methods (Aljundi et al., 2019; Chrysakis & Moens, 2020; Liang & Li, 2023; Sun et al., 2022) use external storage to retain a subset of past data for rehearsal. The emergence of large-scale pre-trained models (Devlin et al., 2019; Dosovitskiy, 2020; He et al., 2022) has shifted the focus of continual learning toward leveraging these powerful backbones. While some approaches fine-tune the entire model (Boschini et al., 2022; Zheng et al., 2023), this is often resource-intensive and prone to forgetting. As a result, recent research explores PEFT in the continual learning setting, offering a more scalable and

memory-efficient alternative. These methods primarily fall into three categories: prompt-based, adapter-based, and LoRA-based. Prompt-based approaches, such as L2P (Wang et al., 2022c), DualPrompt (Wang et al., 2022b), and CODA-Prompt (Smith et al., 2023), learn task-specific prompts for ViTs, allowing the base model to remain fixed while adapting to new tasks. Adapter-based methods (Gao et al., 2023; Zhou et al., 2024) insert small trainable modules (e.g., between feed-forward layers) into frozen backbones, differing from LoRA which injects low-rank updates directly into attention weight matrices. LoRA-based methods aim to avoid interference through various isolation strategies. O-LoRA (Wang et al., 2023) uses a soft orthogonality loss that is often too weak to prevent conflict. InfLoRA (Liang & Li, 2024) stores and projects onto past task gradients—effective but costly in computation and memory. PLAN (Wang et al., 2025) initializes LoRA's down-projection with sparse orthonormal bases, which may limit representational capacity. CL-LoRA (He et al., 2025) allocates separate LoRAs per task, increasing inference overhead. In contrast, SABER does not seek to avoid representational overlap. Instead, it explicitly manages inevitable conflicts through subspace alignment and balanced knowledge recomposition.

## 3. Preliminaries

### 3.1. Problem Definition

Continual Learning aim to learn a model that is sequentially trained on $T$ tasks by a disjoint set of classes $\{C_t\}_{t=1}^T$, where each $C_t \cap C_s = \emptyset$ for $t \neq s$. During a training task $t$, the model only has access to the data from the current task $D_t = \{(\mathbf{x}, \mathbf{y}) | \mathbf{y} \in C_t\}$, which is not allowed to store any exemplars from previous tasks. The final goal is to train a model that can effectively distinguish all classes from both previously encountered tasks (old classes) and the current task (new classes). We follow existing continual learning methods (Liang & Li, 2024; He et al., 2025) to train with a pretrain Vision Transformer (Dosovitskiy, 2020) based on PEFT. We assume the model is $g(f(\cdot))$ where $f(\cdot)$ is the pre-trained ViT backbone and the $g(\cdot)$ is the classifier. The cross entropy loss for training $t$-th task is defined as follows

$$\mathcal{L}_{CE} = -\mathbb{E}_{(\mathbf{x}, \mathbf{y}) \sim D_t} \left[ \log \frac{\exp(g(f(\mathbf{x}))_{\mathbf{y}})}{\sum_{j \in C_t} \exp(g(f(\mathbf{x}))_j)} \right]. \quad (1)$$

### 3.2. Low-Rank Adaptation

Low-Rank Adaptation (LoRA) [20] is an efficient technique for fine-tuning pre-trained models through the decomposition of weight updates into low-rank matrices. Specifically, given a pre-trained weight matrix $W_0 \in \mathbb{R}^{d \times d}$, LoRA expresses its update via a pair of rank-decomposed matrices: a down-projection matrix $A \in \mathbb{R}^{r \times d}$ and an up-projection matrix $B \in \mathbb{R}^{d \times r}$, where the rank $r$ satisfies $r \ll d$. This

approach reduces the number of trainable parameters to only $2 \times r \times d$, thereby improving learning efficiency. For $h = W_0\mathbf{x}$, the modified forward pass for the input $\mathbf{x}$ is given by:

$$h = W_0\mathbf{x} + W\mathbf{x}, \quad \text{where} \quad W = BA. \quad (2)$$

## 4. Method

In this section, we present SABER method (illustrated in Figure 2). To apply LoRA as adapters in continual learning, we introduce task-specific low-rank matrices $\{A_t, B_t\}$ for each task $t$ and adds a new LoRA branch for each task, consisting of the down-projection matrix $A_t \in \mathbb{R}^{r \times d}$ and the up-projection matrix $B_t \in \mathbb{R}^{d \times r}$. During the training of task $t$, only the task-specific matrices $A_t$ and $B_t$ are updated via gradient descent, while the pre-trained weights $W_0$ and all previous task matrices $\{A_i, B_i\}_{i=1}^{t-1}$ remain frozen. The forward pass of the linear layer for $t$-th task for the training stage then becomes:

$$h = W_0\mathbf{x} + B_t A_t\mathbf{x}. \quad (3)$$

At the test stage of $T$-th task, since the task id is unknown, we merge all the LoRA metrics as

$$W = W_0 + \Phi(\{W_t\}_{t=1}^T), \quad (4)$$

where $W_t = B_t A_t$, $\Phi(\cdot)$ stands for the aligment and recomposition manipulation.

### 4.1. Subspace Alignment

Existing LoRA-based CL methods (Wang et al., 2023; Liang & Li, 2024; Wang et al., 2025) perform merging among all $W_t \in \mathbb{R}^{d \times d}$ directly in the unaligned LoRA space, which means the update directions from different tasks are unaligned, leading to severe parameter interference. In our approach, both input-subspace (defined by $A_t$) and output-subspaces (defined by $B_t$) are aligned to a common reference basis to ensure compatibility and comparability across tasks. We further enforce orthogonality within the aligned input-subspace by orthogonalization initialization, which isolates the features that each task processes, preventing interference at the source.

**Input-subspace alignment.** For each $W_t = B_t A_t$ from the $t$-th task, $A_t$ serves as the dimensionality reduction matrix, defines the input-subspace, while $B_t$ governs the output-subspace. As noted in (Zhu et al., 2024), randomly initialized and frozen $A_t$ can achieve performance close to that of a fully fine-tuned version. Based on this finding, we employ orthogonal initialization for each $A_t$, keep all $A_t$ fixed during training, and only update $B_t$. As a foundational step for our subspace alignment, to ensure that all input-subspaces are aligned to a common basis from initialization,

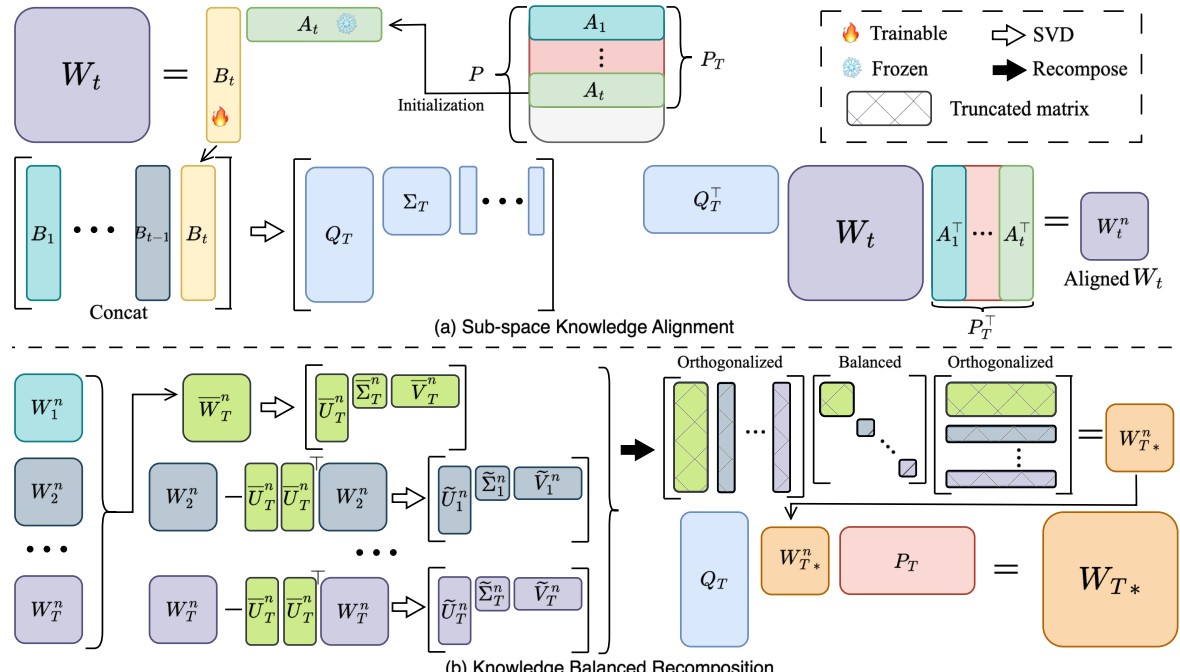

*Figure 2.* The SABER pipeline. (a) Subspace Knowledge Alignment projects all task parameters $W_t$ into a unified subspace via shared orthogonal bases. (b) Balanced Recomposition decomposes the aligned knowledge into task-common and task-specific components, balances their influences, and recomposes them into the weight.

we generate a random orthogonal matrix $P \in \mathbb{R}^{d \times d}$ at the start, and then sequentially select $r$ distinct rows from $P$ to construct each $A_t \in \mathbb{R}^{r \times d}$ as follows

$$A_t = P_{[(t-1) \times r:t \times r]}, \qquad (5)$$

which ensures that all $A_t$ remain row-wise orthogonal to one another and share the same vector basis $P_T$ in the same latent space defined by $P_T = P_{[1:T \cdot r]}$ (i.e., the first $T \times r$ rows of $P$), thereby enforcing a shared reference frame across the input-subspaces of all $\{W_t\}_{t=1}^T$.

**Output-subspace alignment.** Although all $A_t$ matrices now reside in the same input-subspace spanned by $P_t$ and are mutually row-orthogonal, the $B_t$ matrices cannot be subjected to the same property as they must remain trainable. Existing ensuring strict orthogonality among the training of $B_t$ matrices is particularly challenging, as it would conflict with their role in generating task-specific output representations that may need to share common features or semantic concepts. Therefore, we focus solely on aligning $B_t$ to a common latent space. We first find a shared basis for all $B_t$. We concate all $B_t$ horizontally and then perform Singular Value Decomposition (SVD) on this aggregated matrix to obtain

$$B_{stack} = [B_1; B_2; \cdots; B_t] = Q_T \Sigma_T R_T^\top, \qquad (6)$$

where $Q_T \in \mathbb{R}^{d \times (T \cdot r)}$ is the left singular vector of $B_{stack}$ and is a column-orthogonal matrix. The column space of

$Q_T$ spans the union of the column spaces of all the individual $B_t$ and forms a shared basis, thus creating a common coordinate system for all tasks. We set $Q_T$ as the shared basis of the output space of $\{W_t\}_{t=1}^T$. Therefore, we can align all $W_t$ into a common subspace spanned by the reference bases $(P_T, Q_T)$ by

$$W_t^n = Q_T^\top W_t P_T^\top, \qquad (7)$$

where $Q_T^\top Q_T = I$, $P_T P_T^\top = I$, and $W_t^n \in \mathbb{R}^{(T \cdot r) \times (T \cdot r)}$ are the new coordinates in the subspace. Thereby, all $W_t$ are aligned from original space for $W_t \in \mathbb{R}^{d \times d}$ into the common subspace with $W_t^n \in \mathbb{R}^{(T \cdot r) \times (T \cdot r)}$, $\forall t = 1, \cdots, T$, establishing the foundation for subsequent knowledge recomposition. See Appendix A for further analysis and Appendix B for the theoretical necessity of subspace alignment.

### 4.2. Knowledge Recomposition

After aligning all $\{W_t\}_{t=1}^T$ into the same subspace, to integrate knowledge from all tasks, a straightforward approach is to sum all LoRA modules

$$\overline{W}_T^n = \sum_{t=1}^T W_t^n. \qquad (8)$$

However, viewing $\overline{W}_t^n$ from the perspective of singular values and decompose LoRA into directions and the magnitude (i.e., energy) along that direction, reveals two critical issues

with this method: (a) **Amplification of common knowledge.** Directions common across tasks reinforce each other during summation. Their magnitudes accumulate, causing this common knowledge to dominate the merged model and overshadow other information. (b) **Interference of specific knowledge.** Directions unique to individual tasks often point in different or conflicting ways. Their summation leads to destructive interference, effectively turning crucial task-specific knowledge into noise. Consequently, the merged matrix is not a true integration of all knowledge. Instead, it represents a combination of over-amplified common knowledge and a noisy amalgamation of compromised task-specific knowledge.

Based on the above analysis, we separate common and task-specific knowledge per task and recompose them to preserve each task's specificity.

**Task-common knowledge extraction.** We leverage the property that the $\overline{W}_T^n$ tends to amplify common knowledge for its extraction. Specifically, we treat the task-common knowledge as originating from a virtual 0-th task, perform SVD on $\overline{W}_T^n$, and retain only the top-k singular vectors and singular values as the task-common knowledge. Assuming the rank of $\overline{W}_T^n$ is $l$, the task-common knowledge is

$$U_0^n = (\overline{U}_T^n)_{1:k}, \quad \sigma_0^n = \mathrm{diag}(\overline{\Sigma}_T^n)_{1:k}, \quad V_0^n = (\overline{V}_T^n)_{1:k}, \tag{9}$$

where $\overline{W}_T^n = \overline{U}_T^n \overline{\Sigma}_T^n (\overline{V}_T^n)^\top$, $U_0^n \in \mathbb{R}^{Tr \times k}$, $\sigma_0^n \in \mathbb{R}^k$, $V_0^n \in \mathbb{R}^{Tr \times k}$. We set the hyperparameter $\alpha$ to define the proportion to preserve the task-common knowledge as

$$\alpha = \frac{k}{\mathrm{rank}(\overline{W}_T^n)} = \frac{k}{l} \in [0,1]. \tag{10}$$

**Task-specific knowledge extraction.** To extract task-specific knowledge, we first project each $W_t^n$ onto the subspace spanned by $U_0^n$ and subtract this projection to obatin task-specific knowledge as follows

$$\widetilde{W}_t^n = W_t^n - U_0^n U_0^{n\top} W_t^n. \tag{11}$$

Since $\widetilde{W}_t^n$ contains both task-specific information and potential noise, we perform SVD on this residual. We retain the top-$m$ singular vectors and values as the purified task-specific knowledge and ensure the recomposed matrix has the same rank as $\overline{W}_T^n$ with defining $m = (l - k)/T$ as follows

$$U_t^n = (\widetilde{U}_t^n)_{1:m}, \quad \sigma_t^n = \mathrm{diag}(\widetilde{\Sigma}_t^n)_{1:m}, \quad V_t^n = (\widetilde{V}_t^n)_{1:m} \tag{12}$$

where $\widetilde{W}_t^n = \widetilde{U}_t^n \widetilde{\Sigma}_t^n (\widetilde{V}_t^n)^\top$, $U_t^n \in \mathbb{R}^{Tr \times m}$, $\sigma_t^n \in \mathbb{R}^m$, $V_t^n \in \mathbb{R}^{Tr \times m}$. We then combine the task-common and task-specific knowledge by concatenating as

$$U_{ct}^n = [U_0^n; U_1^n; \cdots; U_t^n] \in \mathbb{R}^{Tr \times Tr},$$
$$V_{ct}^n = [V_0^n; V_1^n; \cdots; V_t^n] \in \mathbb{R}^{Tr \times Tr}. \tag{13}$$

To further minimize direction conflicts among the reorganized knowledge components, we orthogonalize both $U_{ct}^n$ and $V_{ct}^n$. This is achieved by finding the closest orthogonal matrix to the original matrix through a structured optimization process, which is known as the orthogonal procrustes problem (Gargiulo et al., 2025). We take $U_{ct}^n$ as an example, the formal definition of the problem is as

$$\min ||U_{ct}^n - U_{ct*}^n||_F^2, \quad s.t.\ U_{ct*}^{n\top} U_{ct*}^n = I, \tag{14}$$

where $|| \cdot ||_F$ denotes the Frobenius norm that represents the distance between the original matrix and the orthogonalized matrix. Taking $U_{ct}^n$ as an example, we can simply compute the SVD of $U_{ct}^n = U_U \Sigma_U V_U^\top$, and obtain the orthogonalized $U_{ct*}^n$, and the same applies to $V_{ct}^n$ as

$$U_{ct*}^n = U_U V_U^\top, V_{ct*}^n = U_V V_V^\top. \tag{15}$$

Thus, we can obtain the orthogonalized $U_{ct*}^n$ and $V_{ct*}^n$. See Appendix C for further analysis of the orthogonalization.

### 4.3. Knowledge Balancing and Reconstruction

During recomposition, both task-common and task-specific directions are preserved, but differing singular value magnitudes cause some tasks to dominate while others degrade. To ensure equitable contribution from all knowledge components, we introduce a knowledge balancing procedure. Given the $l$ singular values in the recomposed matrix ($k$ from task-common knowledge, $m = (l - k)/T$ equally divided among task-specific components). From Eq. (9) and (12), we compute the value of each knowledge direction from the task-common knowledge $\{\sigma_{0,i}^n\}_{i=1}^k$ and use it to proportionally scale the corresponding singular values $\{\sigma_{t,i}^n\}_{i=1}^m$ in the task-specific knowledge of the $t$-th task as

$$\sigma_{t,i_*}^n = \frac{\sigma_{t,i}^n}{\sum_{j=1}^m \sigma_{t,j}^n} \cdot \frac{l-k}{T \cdot k} \sum_{i=1}^k \sigma_{0,i}^n, \tag{16}$$

which ensures the balanced influence across tasks, and preserved internal structure within task-specific knowledge from each task (full derivation in Appendix D). Thus, we reconstruct the LoRA matrix from $U_{ct*}^n$, $V_{ct*}^n$, and the concated balanced $\Sigma_{ct*}^n$. We map it back to the $\mathbb{R}^{d \times d}$ space through an inverse operation by $Q_t$ and $P_t$ from Eq. (7) as follows

$$\Sigma_{ct*}^n = \mathrm{Diag}(\sigma_0^n; \sigma_{1*}^n; \cdots; \sigma_{t*}^n) \in \mathbb{R}^{l \times l},$$
$$W_{T*}^n = U_{ct*}^n \Sigma_{ct*}^n V_{ct*}^{n\top} \tag{17}$$
$$W_{T*} = Q_t U_{ct*}^n \Sigma_{ct*}^n V_{ct*}^{n\top} P_t.$$

Finally, the LoRA matrix of $T$ tasks is

$$W_T = W_0 + W_{T*}. \tag{18}$$

We summarize the whole process of our proposed SABER method in Algorithm 1.

*Table 1.* Comparison of different methods on ImageNet-R and DomainNet. We also report the trainable parameters (%) of each method relative to the pre-trained model. All results are averaged over 5 runs with mean ± standard deviation. **Best** and Second Best results are highlighted.

| Method | Param (%) | ImageNet-R ($T=5$) | | ImageNet-R ($T=10$) | | ImageNet-R ($T=20$) | | DomainNet ($T=5$) | |
|---|---|---|---|---|---|---|---|---|---|
| | | $A_{last}\uparrow$ | $A_{avg}\uparrow$ | $A_{last}\uparrow$ | $A_{avg}\uparrow$ | $A_{last}\uparrow$ | $A_{avg}\uparrow$ | $A_{last}\uparrow$ | $A_{avg}\uparrow$ |
| L2P (2022c) | 0.2 | 72.58 (±0.24) | 77.63 (±0.34) | 70.22 (±0.29) | 76.32 (±0.20) | 67.98 (±0.38) | 74.47 (±0.44) | 69.99 (±0.17) | 76.05 (±0.24) |
| Dual-Prompt (2022b) | 0.5 | 68.90 (±0.69) | 72.88 (±0.71) | 64.73 (±0.58) | 71.18 (±0.79) | 63.60 (±0.84) | 70.02 (±0.77) | 69.61 (±0.25) | 75.36 (±0.36) |
| CODA-Prompt (2023) | 4.6 | 73.03 (±0.65) | 76.90 (±0.40) | 71.28 (±0.31) | 76.22 (±0.60) | 66.88 (±0.65) | 72.19 (±0.25) | 73.30 (±0.16) | 78.75 (±0.08) |
| EASE (2024) | 1.4 | 77.12 (±0.20) | 81.36 (±0.19) | 75.97 (±0.43) | 81.73 (±0.38) | 74.50 (±0.47) | 81.02 (±0.39) | 66.43 (±0.45) | 71.63 (±0.34) |
| InfLoRA (2024) | 0.3 | 76.97 (±0.36) | 82.10 (±0.61) | 75.28 (±0.24) | 80.25 (±0.39) | 70.80 (±0.20) | 77.32 (±0.36) | 74.16 (±0.23) | 79.60 (±0.32) |
| PLAN (2025) | 0.3 | 77.79 (±0.24) | 81.93 (±0.63) | 75.25 (±0.42) | 80.41 (±0.56) | 71.06 (±0.42) | 77.93 (±0.56) | 72.12 (±0.16) | 77.52 (±0.37) |
| CL-LoRA (2025) | 0.3 | 79.23 (±0.12) | 84.76 (±0.08) | 79.64 (±0.40) | 84.79 (±0.40) | 76.70 (±0.27) | 83.69 (±0.36) | 71.94 (±0.23) | 77.49 (±0.17) |
| **SABER (ours)** | 0.3 | **81.85** (±0.28) | **86.49** (±0.19) | **80.30** (±0.29) | **85.98** (±0.27) | **77.77** (±0.18) | **84.01** (±0.39) | **75.34** (±0.28) | **80.12** (±0.20) |

*Table 2.* Comparison of different methods on CIFAR-100 and ImageNet-A. We also report the trainable parameters (%) of each method relative to the pre-trained model. All results are averaged over 5 runs with mean ± standard deviation. **Best** and Second Best results are highlighted.

| Method | Param (%) | CIFAR-100 ($T=5$) | | CIFAR-100 ($T=10$) | | CIFAR-100 ($T=20$) | | ImageNet-A ($T=10$) | |
|---|---|---|---|---|---|---|---|---|---|
| | | $A_{last}\uparrow$ | $A_{avg}\uparrow$ | $A_{last}\uparrow$ | $A_{avg}\uparrow$ | $A_{last}\uparrow$ | $A_{avg}\uparrow$ | $A_{last}\uparrow$ | $A_{avg}\uparrow$ |
| L2P (2022c) | 0.2 | 86.23 (±0.32) | 90.76 (±0.16) | 84.78 (±0.22) | 89.12 (±0.23) | 77.30 (±0.58) | 84.55 (±0.68) | 44.11 (±0.75) | 52.47 (±0.30) |
| Dual-Prompt (2022b) | 0.5 | 86.47 (±0.44) | 90.60 (±0.28) | 84.56 (±0.61) | 89.94 (±0.41) | 79.44 (±0.11) | 86.92 (±0.32) | 39.43 (±0.86) | 52.15 (±0.69) |
| CODA-Prompt (2023) | 4.6 | 88.73 (±0.46) | 92.65 (±0.34) | 86.41 (±0.26) | 91.03 (±0.73) | 79.45 (±0.92) | 86.64 (±0.35) | 48.32 (±0.12) | 59.73 (±0.09) |
| EASE (2024) | 1.4 | 89.36 (±0.60) | 93.03 (±0.18) | 87.59 (±0.68) | 92.13 (±0.38) | 85.32 (±0.76) | 91.55 (±0.28) | 57.47 (±0.64) | 68.00 (±0.43) |
| InfLoRA (2024) | 0.3 | 89.66 (±0.20) | 93.24 (±0.24) | 85.45 (±0.29) | 90.73 (±0.35) | 81.18 (±0.64) | 86.99 (±0.37) | 47.99 (±0.69) | 61.59 (±0.60) |
| CL-LoRA (2025) | 0.3 | 88.97 (±0.23) | 92.66 (±0.20) | 86.68 (±0.35) | 91.48 (±0.33) | 84.63 (±0.25) | 90.67 (±0.32) | 58.79 (±0.54) | 69.24 (±0.49) |
| **SABER (ours)** | 0.3 | **89.98** (±0.37) | **93.67** (±0.20) | **88.06** (±0.22) | **92.51** (±0.25) | **85.64** (±0.45) | 91.47 (±0.39) | **59.82** (±0.31) | **70.06** (±0.21) |

---

**Algorithm 1** SABER

1: **Input:** The data of different tasks $\{\mathcal{D}_t\}_{t=1}^{T}$, a pre-trained ViT model $f(\cdot)$, hyperparemeter $\alpha$.
2: **Output:** The learned LoRA parameters $W_{T*}$.
3: Initialized a random orthogonal matrix $P$
4: **for** $t$ in $1, \cdots, T$ **do**
5:     Expand a new branch for the $t$-th task.
6:     Initialize $A_t$ with $P_{[t \times r:(t+1) \times r]}$ through Eq. (5).
7:     **for** $\mathcal{B}_t$ sampled from $\mathcal{D}_t$ **do**
8:         Update $B_t$ with the loss $\mathcal{L}_{CE}$ from Eq. (1) through gradient descent.
9:     **end for**
10: **end for**
11: Obtain $Q_T$ from $\{B_t\}_{t=1}^{T}$ through Eq. (6).
12: Align $\{W_t\}_{t=1}^{T}$ into $\{W_t^n\}_{t=1}^{T}$ through Eq. (7).
13: Obtain task-common knowledge through Eq. (8) and Eq. (9).
14: Obtain task-specific knowledge through Eq. (11) and Eq. (12).
15: Recompose all knowledge through Eq. (13), and orthogonalize the direction of knowledge through Eq. (15).
16: Balance all knowledge through Eq. (16), and reconstruct $W_{T*}$ through Eq. (17).

## 5. Experiments

### 5.1. Experimental Setting

**Datasets.** We conduct comprehensive experiments on four widely recognized CL benchmarks, including CIFAR-100 (Krizhevsky, 2009), ImageNet-R (Hendrycks et al., 2021a), ImageNet-A (Hendrycks et al., 2021b), and DomainNet (Peng et al., 2019). CIFAR-100 contains 100 natural object classes, ImageNet-R and ImageNet-A each contain 200

classes selected from ImageNet (Russakovsky et al., 2015). DomainNet features 345 classes spanning six distinct domains. For all datasets, we create different task sequences by dividing the classes into $T$ task with equal size (e.g., $T = 10$ tasks with 10 classes each for CIFAR-100). Specifically, we divide DomainNet into 5 tasks, each with 69 classes.

**Baselines and evaluation metrics.** The performance of SABER is compared against several state-of-the-art CL methods, including prompt-based methods L2P (Wang et al., 2022c), DualPrompt (Wang et al., 2022b), CODA-Prompt (Smith et al., 2023), and adapter-based method EASE (Zhou et al., 2024), and LoRA-based methods InfLoRA (Liang & Li, 2024), PLAN (Wang et al., 2025), and CL-LoRA (He et al., 2025). Following the standard evaluation protocol, we compare the average accuracy $A_{avg} = \frac{1}{T} \sum_{t=1}^{T} Acc_t$, and the final accuracy $A_{last}$.

**Implementation details.** We adopt ViT-B/16 (Dosovitskiy, 2020) with $N = 12$ Transformer blocks pre-trained on ImageNet-21K (Deng et al., 2009) as our backbone across all experiments. We also evaluate our method on the self-supervised ViT-B/16 (pre-trained by iBOT-1k (Zhou et al., 2022)) to verify its effectiveness across training paradigms (see Appendix F). We use rank $r = 10$ in all LoRA modules, which are inserted into the Multi-Head Self-Attention layers on query ($W_q$) and value ($W_v$) projection matrices. The implementation of existing methods are based on the LAMDA-PILOT (Sun et al., 2025) or their official code. For PLAN (Wang et al., 2025), we take the results reported in

*Table 3.* Ablation study of our SABER method for three component subspace alignment (SA), knowledge recomposition (KR), and knowledge balancing (KB).

| Method | ImageNet-R ($T = 5$) | | ImageNet-R ($T = 10$) | | ImageNet-R ($T = 20$) | |
|---|---|---|---|---|---|---|
| | $A_{last} \uparrow$ | $A_{avg} \uparrow$ | $A_{last} \uparrow$ | $A_{avg} \uparrow$ | $A_{last} \uparrow$ | $A_{avg} \uparrow$ |
| Baseline | 72.27 (±0.43) | 78.63 (±0.32) | 67.65 (±0.39) | 75.33 (±0.25) | 62.98 (±0.76) | 70.14 (±0.54) |
| w/ orthogonal init. $A_t$ | 74.83 (±0.42) | 79.76 (±0.18) | 72.39 (±0.62) | 76.10 (±0.21) | 67.58 (±0.56) | 74.83 (±0.59) |
| w/ SA | 75.48 (±0.44) | 80.08 (±0.25) | 73.25 (±0.53) | 76.89 (±0.31) | 68.41 (±0.27) | 75.09 (±0.33) |
| w/ KR | 76.86 (±0.43) | 81.37 (±0.48) | 74.08 (±0.47) | 78.28 (±0.49) | 69.63 (±0.24) | 76.02 (±0.37) |
| w/ KR+KB | 77.64 (±0.28) | 82.72 (±0.79) | 75.11 (±0.14) | 80.59 (±0.33) | 71.44 (±0.84) | 77.43 (±0.88) |
| w/ SA+KR | 80.81 (±0.27) | 84.29 (±0.21) | 78.53 (±0.32) | 83.18 (±0.52) | 74.78 (±0.25) | 81.86 (±0.34) |
| **SABER** | **81.85** (±0.28) | **86.49** (±0.19) | **80.30** (±0.29) | **85.98** (±0.27) | **77.77** (±0.18) | **84.01** (±0.39) |

their paper since they have not released the code. We use a fixed hyperparameter with $\alpha = 0.6$. See Appendix G for training details.

## 5.2. Experimental Result

As shown in Table 1 and 2, SABER achieves competitive results across different tasks $T$ on multiple benchmarks. (See Appendix F for performance curves for each task) Compared to prompt-based methods, our approach achieves significant performance improvements while maintaining high parameter efficiency. Notably, our method requires less than one-tenth of the trainable parameters required by CODA-Prompt (Smith et al., 2023). When evaluated against EASE (Zhou et al., 2024), which incorporates additional adapters into MLP layers, our method achieves comparable performance across varying numbers of tasks on CIFAR-100 with fewer parameters, and demonstrates even more substantial advantages on the other three datasets. Compared to other LoRA-based methods (Liang & Li, 2024; Wang et al., 2025; He et al., 2025), our approach achieves improved performance advantages on the four dataset under different task quantity scenarios. This demonstrates the generality of our approach. By effectively separating and reorganizing knowledge components, SABER fosters better collaboration and retention, leading to competitive performance.

## 5.3. Abaltion Study

**Effect of different components.** We conduct ablation studies to validate the effectiveness of key components in SABER, including the subspace alignment (**SA**) illustrated in Section 4.1, knowledge recomposition (**KR**) illustrated in Section 4.2, and knowledge balancing (**KB**) illustrated in Section 4.3. The baseline method trains a separate LoRA module for each task and integrates it directly into the model upon task completion. We further introduce orthogonal initialization of $A_t$ as an enhanced baseline, which already brings noticeable improvement by reducing interference among tasks. We keep the matrices $\{A_t\}_{t=1}^{T}$ frozen and exclusively train the matrix $B_t$ across our entire experimental

suite. Notably, since KB operates within the KR process, it cannot be used alone and must be employed jointly with KR.

As shown in Table 3, our analysis reveals profound synergistic effects between the components. SA establishes a structured foundation for knowledge operations by constructing a shared orthogonal subspace. The KR module then enables effective separation and recombination of task-common and task-specific knowledge within this aligned space. Notably, while KR alone delivers substantial gains, its combination with SA produces a remarkable performance leap, demonstrating that the aligned subspace significantly enhances the precision of knowledge separation. KB further refines this process by operating as an internal regulatory mechanism within KR, balancing task-specific knowledge weights to prevent dilution during recomposition.

The synergy between SA and KR becomes particularly crucial in long task sequences (e.g., T=20), where their combined operation leads to notable performance gains. This indicates that the aligned representation space not only mitigates task interference but, more importantly, provides a geometrically consistent working environment for knowledge recombination operations. The incorporation of KB further optimizes this process by ensuring that unique knowledge from each task is adequately preserved and utilized. When all three components are fully integrated, SABER achieves optimal performance across all task sequence lengths, validating the tight complementary relationship between the components: SA lays the foundation for KR, while KB refines KR's output, forming a synergistically enhanced framework. This progressive performance improvement pattern convincingly demonstrates the internal coherence of SABER's design and its capability to maintain long-term stability in continual learning scenarios.

**Different initialization of $A_t$.** Table 4 compares different initialization strategies for the frozen matrix $A_t$, evaluated on ImageNet-R ($T = 10$). We assess three properties: self-orthogonality ($A_i \perp A_i$), mutual orthogonality ($A_i \perp A_j$), and non-sparsity. Random initialization lacks orthogonality,

causing severe task interference and catastrophic forgetting. Intra-Ortho (He et al., 2025; Liang & Li, 2024) enforces self-orthogonality but not alignment across tasks, leading to residual interference during knowledge recomposition. The standard basis (Wang et al., 2025) satisfies both orthogonality conditions and ensures a shared input space, yet its extreme sparsity hinders learning capacity. In contrast, SABER constructs $A_t$ from dense random orthogonal matrices with non-overlapping rows, achieving mutual orthogonality, subspace alignment, and strong representational expressivity simultaneously.

*Table 4.* Comparison of different initialization methods of $A_t$ on ImageNet-R ($T = 10$).

| Method \ Property | $A_i \perp A_i$ | $A_i \perp A_j$ | Non-sparse | $A_{last} \uparrow$ | $A_{avg} \uparrow$ |
|---|---|---|---|---|---|
| Random init. | | | $\checkmark$ | 2.76 (±1.34) | 25.35 (±5.68) |
| Intra-Ortho. init. | $\checkmark$ | | $\checkmark$ | 72.39 (±0.62) | 76.10 (±0.21) |
| Standard basis | $\checkmark$ | $\checkmark$ | | 76.29 (±0.45) | 81.43 (±0.33) |
| **Ours** | $\checkmark$ | $\checkmark$ | $\checkmark$ | **80.30 (±0.29)** | **85.98 (±0.27)** |

**Abaltion study on $\alpha$.** The hyperparameter $\alpha$ represents the ratio of singular values retained from the task-common knowledge matrix $\overline{W}_T^n$, thereby determining the proportion of task-common knowledge incorporated into the final parameter matrix. We empirically evaluated the impact of different values of $\alpha$ on average accuracy $A_{avg}$ on ImageNet-R ($T = 10$). As illustrated in the Figure 3, the optimal performance is achieved when $\alpha = 0.6$, which highlights the importance of using task-specific knowledge. When $\alpha > 0.8$, an excessively high proportion of task-common knowledge leads to performance degradation, as the model underutilizes task-specific knowledge. In the extreme case where $\alpha = 1.0$, no task-specific knowledge is incorporated, and SABER simplifies to its SA component alone. Conversely, when $\alpha$ is too small, performance also declines due to an over-reliance on task-specific knowledge and an insufficient use of task-common knowledge. Furthermore, SABER is robust to $\alpha$ within $[0.5, 0.8]$, accuracy drops by at most 0.63%. See Appendix F for additional experimental reuslts.

## 5.4. Analysis of Computational Complexity

We compared the total memory requirements, training time, and the average final-task inference time of different methods under ImageNet-R ($T = 10$) setting. As summarized in Table 5, among the three categories of methods, prompt-based methods (ie, L2P (Wang et al., 2022c), Dual-Prompt (Wang et al., 2022b), CODA-Prompt (Smith et al., 2023)) excel in memory usage and inference speed but can be constrained in performance on more complex tasks. The adapter-based method EASE (Zhou et al., 2024) creates a new adapter for each task and utilizes all of them during inference, which incurs a substantial cost, particularly in in-

ference time (115.0s). While LoRA-based methods are generally efficient. InfloRA (Liang & Li, 2024) demands high memory (24.33 MB) since it requires storing the gradients of past samples. CL-LoRA (He et al., 2025) suffers from slow inference (110.8s) since it cannot integrate all the LoRA modules into the model after training. SABER achieves competitive performance with low training/inference time and acceptable storage, offering well-balanced practical efficiency. See Appendix E for further complexity analysis.

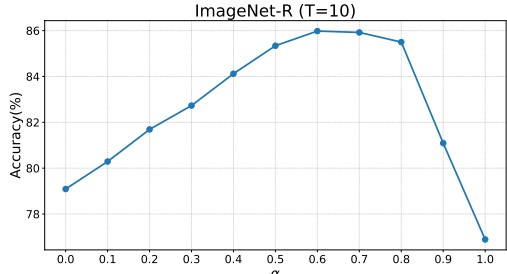

*Figure 3.* Comparison of different $\alpha$ on ImageNet-R ($T = 10$).

*Table 5.* Comparison of computational efficiency, including memory usage, training time, and inference time.

| Method | Memory (MB) | Training (min) | Inference (s) |
|---|---|---|---|
| L2P | 0.31 | 22.08 (±0.47) | 15.4 (±0.6) |
| DualPrompt | 1.82 | 21.06 (±0.51) | 14.6 (±0.6) |
| CODA-Prompt | 14.65 | 115.51 (±0.23) | 15.2 (±1.1) |
| EASE | 7.37 | 31.00 (±0.30) | 115.0 (±0.7) |
| InfloRA | 24.33 | 76.66 (±0.13) | 21.2 (±3.9) |
| PLAN | 2.72 | - | - |
| CL-LoRA | 4.06 | 46.71 (±0.69) | 110.8 (±1.9) |
| SABER | 4.63 | 54.43 (±0.16) | 18.8 (±1.9) |

## 6. Conclusion

In this work, we propose SABER (Subspace-Aligned Balanced Recomposition), a novel continual learning method that offers a complementary perspective from interference avoidance to structured knowledge composition. Unlike existing approaches that either enforce strict orthogonality or rely on modular separation, SABER establishes a unified subspace alignment framework where inherent task conflicts are managed through balanced recomposition. Our method demonstrates that by aligning tasks to a shared latent space, strategically decomposing and recomposing knowledge components, and implementing energy-aware balancing, we can achieve a promising balance of stability-plasticity trade-offs and deliver competitive performance.

**Limitations.** Our method is designed for the offline continual learning setting, and its performance has not been evaluated under online or federated continual learning scenarios.

## Acknowledgements

The work is supported by the National Natural Science Foundation of China (No.62302170), the Guangdong Basic and Applied Basic Research Foundation (No.2024A1515010187), the Guangdong Natural Science Funds for Distinguished Young Scholars (Grant 2023B1515020097), the Singapore Ministry of Education Academic Research Fund Tier 2 (Award No. MOE-T2EP20125-0016), and the Lee Kong Chian Fellowships.

## Impact Statement

This paper presents work whose goal is to advance the field of Machine Learning. There are many potential societal consequences of our work, none which we feel must be specifically highlighted here.

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

# A. Theoretical Guarantees for Subspace Alignment

In this section, we provide a theoretical justification for the subspace alignment strategy employed in SABER. Specifically, we prove that the matrices $P_T$ and $Q_T$, as constructed in Section 4.1 of the main paper, indeed constitute shared bases for the input and output subspaces, respectively. This guarantees that all LoRA modules can be losslessly projected into a common representation space, enabling subsequent knowledge recomposition without discarding essential information.

Consider $T$ tasks, each associated with a LoRA pair $(A_t, B_t)$, where $A_t \in \mathbb{R}^{r \times d}$ and $B_t \in \mathbb{R}^{d \times r}$. The weight update for task $t$ is $W_t = B_t A_t$. The alignment procedure aims to express each $A_t$ and $B_t$ in terms of the shared bases $P_T$ and $Q_T$, i.e.,

$$A_t = M_t P_T, \quad B_t = Q_T N_t, \tag{19}$$

for the core coefficient matrices $M_t$ and $N_t$. We now establish the validity of this decomposition.

## A.1. Shared Basis for the Input-Subspace via $P_T$

From Eq. (5), $\{A_t\}_{t=1}^T$ are instantiated from a fixed random orthogonal matrix $P \in \mathbb{R}^{d \times d}$ satisfying $PP^\top = I_d$. Each $A_t$ is formed by selecting a distinct block of $r$ consecutive rows:

$$A_t = P_{[(t-1) \times r : t \times r, :]}, \quad t = 1, \ldots, T. \tag{20}$$

$\{A_t\}_{t=1}^T$ are frozen during training. The shared basis of the input-subspace $P_T \in \mathbb{R}^{Tr \times d}$ is defined as the matrix comprising the first $Tr$ rows of $P$ :

$$P_T := P_{[1:Tr, :]}. \tag{21}$$

By construction, $A_t$ corresponds exactly to the $t$-th block of $P_T$. Since $P$ is orthogonal, its rows are orthonormal, and thus the rows of $P_T$ satisfy:

$$P_T P_T^\top = I_{Tr}. \tag{22}$$

**Definition A.1** (Joint Input-Subspace). The joint input-subspace $\mathcal{S}_{\text{in}}$ is defined as

$$\mathcal{S}_{\text{in}} := \text{span} \left( \bigcup_{t=1}^T \text{row}(A_t) \right). \tag{23}$$

**Lemma A.2.** *The row space of $P_T$ coincides with $\mathcal{S}_{in}$ :*

$$\text{row}(P_T) = \mathcal{S}_{in}. \tag{24}$$

*Proof.* Every row of each $A_t$ is a row of $P_T$, so $\bigcup_t \text{row}(A_t) \subseteq \text{row}(P_T)$, implying $\mathcal{S}_{\text{in}} \subseteq \text{row}(P_T)$. Conversely, every row of $P_T$ belongs to some $A_t$, so $\text{row}(P_T) \subseteq \mathcal{S}_{\text{in}}$. Hence, equality holds. □

**Theorem A.3** (Lossless Representation via $P_T$). *For each task t, there exists a unique matrix $M_t \in \mathbb{R}^{r \times Tr}$ such that*

$$A_t = M_t P_T, \quad \text{and} \quad \|A_t - M_t P_T\|_F = 0. \tag{25}$$

*Proof.* From Lemma A.2, each row of $A_t$ lies in $\text{row}(P_T)$. Because the rows of $P_T$ are orthonormal (Eq. (22)), the orthogonal projection of any such row onto $\text{row}(P_T)$ is exact. For any row vector $a \in \text{row}(A_t)$, we have $a = a P_T^\top P_T$. Stacking all rows of $A_t$ yields $A_t = A_t P_T^\top P_T$. Defining $M_t := A_t P_T^\top \in \mathbb{R}^{r \times Tr}$, we obtain $A_t = M_t P_T$ with zero reconstruction error. □

This confirms that $P_T$ serves as an exact, non-parametric shared basis for all input projections.

## A.2. Shared Basis for the Output-Subspace via $Q_T$

The output projection matrices $\{B_t\}_{t=1}^{T}$ are learned during fine-tuning. To construct a unified representation, SABER forms the horizontally stacked matrix

$$B_{\text{stack}} := [B_1, B_2, \ldots, B_T] \in \mathbb{R}^{d \times (Tr)}. \tag{26}$$

and computes its compact singular value decomposition:

$$B_{\text{stack}} = Q_T \Sigma_T R_T^\top, \tag{27}$$

where $Q_T \in \mathbb{R}^{d \times k}$ has orthonormal columns ($Q_T^\top Q_T = I_k$), $k = \text{rank}(B_{\text{stack}})$, and $\Sigma_T \in \mathbb{R}^{k \times k}$ contains the positive singular values.

**Definition A.4** (Joint Output-Subspace). The joint output subspace is

$$\mathcal{S}_{\text{out}} := \text{span}\left(\bigcup_{t=1}^{T} \text{col}(B_t)\right) = \text{col}(B_{\text{stack}}). \tag{28}$$

**Lemma A.5.** *The column space of $Q_T$ equals the joint output-subspace:*

$$\text{col}(Q_T) = \mathcal{S}_{out}. \tag{29}$$

*Proof.* By the definition of SVD, the columns of $Q_T$ form an orthonormal basis for $\text{col}(B_{\text{stack}})$. Since $B_{\text{stack}}$ is composed of all $B_t$, its column space is precisely $\mathcal{S}_{\text{out}}$. $\square$

**Theorem A.6** (Lossless Representation via $Q_T$). *For each task t, there exists a matrix $N_t \in \mathbb{R}^{k \times r}$ such that*

$$B_t = Q_T N_t, \quad and \quad \|B_t - Q_T N_t\|_F = 0. \tag{30}$$

*Proof.* From Lemma A.5, $\text{col}(B_t) \subseteq \text{col}(Q_T)$. Because $Q_T$ has orthonormal columns, the least-squares solution to $Q_T N_t = B_t$ is $N_t = Q_T^\top B_t$. Substituting gives $Q_T N_t = Q_T Q_T^\top B_t$. The matrix $Q_T Q_T^\top$ is the orthogonal projector onto $\text{col}(Q_T)$, and since $B_t$ already lies in this subspace, the projection is identity: $Q_T Q_T^\top B_t = B_t$. Hence, equality holds exactly. $\square$

Together, Theorems A.3 and A.6 establish that the subspace alignment mechanism in SABER, which leverages $P_T$ for the input-subspace basis and $Q_T$ for the output-subspace basis, provides an exact and information-preserving common representation for all LoRAs. This shared basis ensures that no knowledge is lost during subspace alignment, thereby facilitating subsequent knowledge recomposition operations in the aligned subspace.

# B. The Necessity of Subspace Alignment for Knowledge Recomposition

In this section, we establish that subspace alignment is a necessary prerequisite for effective knowledge recomposition in SABER. We justify this necessity from two perspectives: (1) the representation ambiguity of unaligned low-rank updates, and (2) the signal-to-noise ratio (SNR) gain achieved through dimensionality reduction.

## B.1. Representation Ambiguity in Unaligned LoRA Updates

Each task's update is parameterized as $W_t = B_t A_t$, where $A_t \in \mathbb{R}^{r \times d}$ and $B_t \in \mathbb{R}^{d \times r}$. However, this factorization is not unique. For any invertible matrix $R \in \mathbb{R}^{r \times r}$, the following holds:

$$B_t A_t = (B_t R)(R^{-1} A_t). \tag{31}$$

In the worst case, two tasks that learn identical weight changes ($W_i = W_j$) may produce $B_i$ and $B_j$ with orthogonal column spaces(i.e., $\text{col}(B_i) \perp \text{col}(B_j)$). This representation ambiguity implies that raw LoRA matrices lack a common semantic coordinate system. Directly applying operations such as summation or singular value decomposition across unaligned updates conflates geometric artifacts with true task knowledge. Therefore, a shared basis is required to make knowledge representations comparable across tasks.

**B.2. Signal-to-Noise Ratio Improvement via Dimensionality Reduction**

Subspace alignment aligns all $W_t$ into a common $Tr$-dimensional subspace (i.e., $W_t^n \in \mathbb{R}^{Tr \times Tr}$) using global bases $P_T$ and $Q_T$. The aligned LoRA matrix for task $t$ is defined as:

$$W_t^n = Q_T^\top W_t P_T^\top. \tag{32}$$

Assume each $W_t$ consists of a true signal with Frobenius norm $\mu$ and isotropic noise matrix $E_t$, which is an i.i.d. zero-mean entries with variance $\nu^2 = \sigma^2/d^2$ and satisfying $\mathbb{E}[\|E_t\|_F^2] = \sigma^2$. In the original $d \times d$ space, the SNR is:

$$\text{SNR}_{\text{raw}} = \frac{\mu^2}{\sigma^2}. \tag{33}$$

After aligning into the $Tr \times Tr$ subspace, the signal is largely preserved because $P_T$ and $Q_T$ are constructed from dominant directions across tasks. Meanwhile, the noise energy is reduced. Since the noise is uniformly distributed over $d^2$ dimensions, its expected energy in the $Tr \times Tr$ subspace scales with the dimension ratio:

$$
\begin{aligned}
\mathbb{E}\big[\|Q_T^\top E_t P_T^\top\|_F^2\big] &= \text{tr}(P_T^\top \mathbb{E}[E_t^\top Q_T Q_T^\top W_t] P_T^\top) \\
&= \nu^2 Tr \cdot \text{tr}(P_T P_T^\top) \\
&= \nu^2 (Tr)^2 \\
&= \left(\frac{Tr}{d}\right)^2 \sigma^2.
\end{aligned}
\tag{34}
$$

Thus, the SNR after alignment becomes:

$$\text{SNR}_{\text{aligned}} \approx \frac{\mu^2}{(Tr/d)^2 \sigma^2} = \left(\frac{d}{Tr}\right)^2 \cdot \text{SNR}_{\text{raw}}. \tag{35}$$

In practical settings, $Tr < d$ (e.g., $d = 768$, $T = 20$, $r = 10$ yields $Tr = 200$), which means $\text{SNR}_{\text{aligned}} \gg \text{SNR}_{\text{raw}}$. This enhanced SNR ensures that subsequent knowledge recomposition operate on high-fidelity knowledge representations.

As summarized, these two points demonstrate that subspace alignment is necessary for reliable knowledge recomposition.

## C. Analysis of the Orthogonalization Process

In the knowledge recomposition stage of SABER, we orthogonalize the concatenated knowledge matrices $U_{ct}^n$ and $V_{ct}^n$ to eliminate directional conflicts among task components. This is formulated as an orthogonal Procrustes problem:

$$\min \|U_{ct}^n - U_{ct*}^n\|_F^2 \quad \text{,s.t.} \quad U_{ct*}^n (U_{ct*}^n)^T = I.$$

The solution is obtained via SVD: if $U_{ct}^n = U_U \Sigma_U V_U^T$, then the closest orthogonal matrix is $U_{ct*}^n = U_U V_U^T$. The same applies to $V_{ct}^n$. Thus, the orthogonalization error is defined as the Frobenius norm of the difference:

$$\mathcal{E}_U = \|U_{ct}^n - U_{ct*}^n\|_F.$$

This error can be expressed in terms of the singular values $\sigma_i$ of $U_{ct}^n$, where $\sigma_1 \geq \sigma_2 \geq \cdots \geq \sigma_l \geq 0$:

$$\mathcal{E}_U = \|\Sigma_U - I\|_F = \sqrt{\sum_{i=1}^{l}(\sigma_i - 1)^2}.$$

From this, we can derive an upper bound:

$$\mathcal{E}_U \leq \sqrt{l} \cdot |\sigma_1 - 1|.$$

And a lower bound:

$$\mathcal{E}_U \geq \sqrt{\sum_{i=1}^{l}(\sigma_i - 1)^2} \geq \sqrt{l} \cdot |\sigma_l - 1|.$$

These bounds confirm that the error is controlled by the deviation of the singular values of $U_{ct}^n$ from unity.

While the above analysis provides theoretical bounds, the orthogonalization error is not a primary concern for the performance of SABER. The key objective of this step is not to preserve the exact original directions of $U_{ct}^n$, but to ensure mutual orthogonality among the final knowledge components. Even if the orthogonalized directions $U_{ct*}^n$ deviate slightly from the original $U_{ct}^n$, what truly matters is that $U_{ct*}^n$ forms an orthogonal basis. Therefore, the orthogonalization error is acceptable. It is the structural property of orthogonality itself, not the directional fidelity, that is fundamental to mitigating interference and enabling stable knowledge recomposition in SABER.

## D. Derivation of the Knowledge Balancing Formula

In this section, we explains how Eq. (16) is derived from the knowledge balancing procedure described in Section 4.3.

Let $\Sigma_{ct}^n$ be the recomposed singular value matrix with $l$ singular values, where $k$ values belong to the task-common component and the remaining $l - k$ are distributed equally among $T$ tasks. Each task thus receives

$$m = \frac{l - k}{T} \tag{36}$$

singular values in its task-specific component. Let $\{\sigma_{0,i}^n\}_{i=1}^{k}$ denote the singular values of the task-common knowledge, and $\{\sigma_{t,i}^n\}_{i=1}^{m}$ denote the singular values of the task-specific knowledge of $t$-th task. The total magnitude of the singular values of the task-common knowledge is

$$M_{\text{common}} = \sum_{i=1}^{k} \sigma_{0,i}^n. \tag{37}$$

Thus, the magnitude per knowledge direction is

$$M_{per} = \frac{M_{\text{common}}}{k} = \frac{1}{k}\sum_{i=1}^{k} \sigma_{0,i}^n. \tag{38}$$

To ensure equitable contribution across tasks, SABER redistributes this total magnitude proportionally to the number of task-specific directions. The balancing procedure assigns a global scaling factor $\lambda$ defined as

$$\lambda = m \cdot M_{per} = \frac{l - k}{T \cdot k}\sum_{i=1}^{k} \sigma_{0,i}^n. \tag{39}$$

This factor can be interpreted as follows: the total common magnitude $M_{\text{common}}$ is first averaged over the $k$ common directions, yielding $M_{per}$. The average magnitude is then multiplied by the number of specific directions per task, $(l - k)/T$, resulting in the total magnitude allocated to each task's specific component.

Each task-specific singular value $\sigma_{t,i}^n$ is then normalized among its task (i.e., among $m$ singular values) and then scaled by this factor to obtain the rebalanced singular value:

$$\sigma_{t,i*}^n = \frac{\sigma_{t,i}^n}{\sum_{j=1}^{m} \sigma_{t,j}^n} \cdot \lambda = \frac{\sigma_{t,i}^n}{\sum_{j=1}^{m} \sigma_{t,j}^n} \cdot \frac{l - k}{T \cdot k}\sum_{i=1}^{k} \sigma_{0,i}^n. \tag{40}$$

Thus, we obtained Eq. (16).

## E. Computational Complexity Analysis

In this Section, we theoretically analyze the computational complexity of the relatively costly SVD operations in SABER. Although SABER requires several SVD operations, its time and storage consumption are still acceptable (see Section 5.4).

Let the LoRA matrix $W \in \mathbb{R}^{d \times d}$ with rank $r$, and let $T$ and $L$ be the number of tasks and layers, respectively. SABER performs:

- One SVD on $B_{stack} \in \mathbb{R}^{T \cdot d \times d}$ per layer (line 11, Algorithm 1) with complexity $\mathcal{O}(TLd^3)$

- One SVD on $\overline{W}_T^n \in \mathbb{R}^{Tr \times Tr}$ per layer (line 13, Algorithm 1) with complexity $\mathcal{O}(T^3 Lr^3)$

- One SVD on each $\widetilde{W}_t^n \in \mathbb{R}^{Tr \times Tr}$ per layer (line 14, Algorithm 1) with complexity $\mathcal{O}(T^4 Lr^3)$

- Two SVD on $U_{ct}^n, V_{ct}^n \in \mathbb{R}^{Tr \times Tr}$ per layer (line 15, Algorithm 1) with complexity $\mathcal{O}(2T^3 Lr^3)$

The total complexity equals:

$$\begin{aligned} \mathcal{O}(SABER) &= \mathcal{O}(TLd^3 + T^3 Lr^3 + T^4 Lr^3 + 2T^3 Lr^3) \\ &= \mathcal{O}(TLd^3 + 3T^3 Lr^3 + T^4 Lr^3) \\ &= \mathcal{O}(TLd^3 + T^4 Lr^3). \end{aligned} \tag{41}$$

To identify the dominant term, we introduce the low-rank ratio

$$\epsilon := \frac{r}{d} \ll 1, \tag{42}$$

which quantifies the degree of parameter reduction enabled by LoRA. The relative magnitude of the two terms is then governed by

$$\frac{T^4 Lr^3}{TLd^3} = T^3 \epsilon^3. \tag{43}$$

In practical settings, LoRA is typically applied with $d = 768$ (e.g., in ViT) and a small adaptation rank such as $r = 10$, yielding $\epsilon \approx 0.013$. Moreover, the number of tasks is usually modest; for instance, $T < 50$ in most offline class-incremental learning scenarios. Under these conditions,

$$T^3 \epsilon^3 \lesssim 50^3 \cdot (0.013)^3 \approx 125{,}000 \cdot 2.2 \times 10^{-6} \approx 0.275 \ll 1, \tag{44}$$

which confirms that the $TLd^3$ term dominates the overall cost. Consequently, the total computational complexity is well approximated by

$$\mathcal{O}(SABER) = \mathcal{O}(TLd^3), \tag{45}$$

indicating that the runtime is primarily determined by the base model dimension $d$, while the low-rank structure ($r \ll d$) effectively suppresses the overhead associated with task-wise interactions even when higher-order coupling terms (e.g., $\mathcal{O}(T^4 Lr^3)$) are present in the full expression.

## F. Additional Experimental Results

In this section, we provide the performance curve of each learning task with different methods on four different datasets: Figure 4 shows the results of ImageNet-R and DomainNet; Figure 5 shows the results of CIFAR-100 and ImageNet-A.

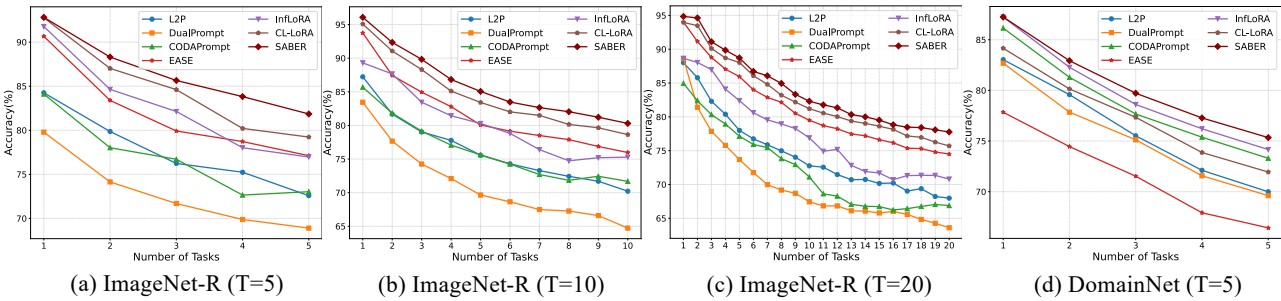

(a) ImageNet-R (T=5)    (b) ImageNet-R (T=10)    (c) ImageNet-R (T=20)    (d) DomainNet (T=5)

*Figure 4.* Comparison of the performance of different methods during the learning of ImageNet-R and DomainNet.

Besides, we evaluated different methods on CIFAR-100 and ImageNet-R ($T = 10$) based on ViT-B/16 model pre-trained with iBOT-1k (Zhou et al., 2022). As shown in Table 6, methods employing self-supervised pre-trained models generally

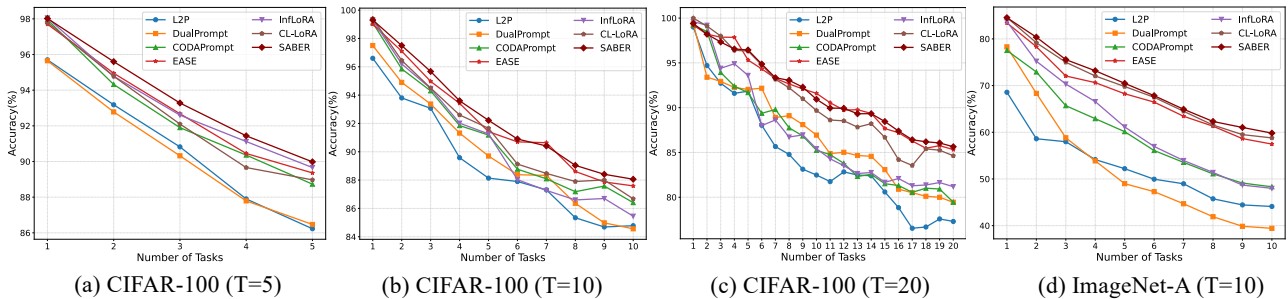

(a) CIFAR-100 (T=5)   (b) CIFAR-100 (T=10)   (c) CIFAR-100 (T=20)   (d) ImageNet-A (T=10)

*Figure 5.* Comparison of the performance of different methods during the learning of CIFAR-100 and ImageNet-A.

*Table 6.* Comparison of different methods on CIFAR-100 and ImageNet-R ($T = 10$) using iBOT-1k self-supervised pre-trained model.

| Method | CIFAR-100 (T = 10) | | ImageNet-R (T = 10) | |
|---|---|---|---|---|
| | $A_{last} \uparrow$ | $A_{avg} \uparrow$ | $A_{last} \uparrow$ | $A_{avg} \uparrow$ |
| L2P (2022c) | 72.80 (±0.35) | 80.58 (±0.28) | 60.78 (±0.38) | 66.88 (±0.16) |
| Dual-Prompt (2022b) | 74.11 (±0.48) | 81.69 (±0.47) | 59.58 (±0.07) | 67.29 (±0.28) |
| CODA-Prompt (2023) | 79.83 (±0.29) | 86.95 (±0.22) | 67.82 (±0.44) | 75.06 (±0.13) |
| EASE (2024) | 79.42 (±0.43) | **87.05** (±0.32) | 69.68 (±0.39) | 76.87 (±0.25) |
| InfLoRA (2024) | 77.71 (±0.29) | 86.03 (±0.09) | 71.94 (±0.43) | 78.47 (±0.32) |
| CL-LoRA (2025) | 80.37 (±0.27) | 86.90 (±0.22) | 73.23 (±0.39) | 79.64 (±0.25) |
| **SABER** | **81.07** (±0.36) | 86.98 (±0.43) | **74.38** (±0.29) | **80.76** (±0.27) |

underperform relative to their supervised counterparts, while SABER demonstrates competitive performance compared to other methods, confirming its robustness even when integrated with self-supervised pre-training model.

We also conducted a sensitivity analysis of $\alpha$ across all benchmarks (Table R2). The results show that SABER performs robustly over a wide range of values. For datasets with higher inter-task similarity (e.g., CIFAR100), larger $\alpha$ performs slightly better, indicating stronger reliance on shared components. For more diverse datasets (e.g., ImageNet-A, DomainNet), a smaller $\alpha$ is preferred, as excessive sharing can introduce mild interference. Across all settings, $\alpha = 0.6$ provides a stable and consistently strong default.

*Table 7.* Accuracy of different datasets and $\alpha$

| Datasets \ $\alpha$ | 0.2 | 0.4 | 0.6 | 0.8 | 1.0 |
|---|---|---|---|---|---|
| CIFAR100 | 90.12 | 91.85 | 92.51 | **92.63** | 89.74 |
| ImageNet-R | 81.69 | 84.12 | **85.98** | 85.50 | 76.89 |
| ImageNet-R(20steps) | 79.24 | 82.37 | **84.01** | 83.85 | 74.12 |
| ImageNet-A | 67.21 | 69.54 | **70.06** | 69.31 | 63.42 |
| DomainNet | 76.83 | 79.67 | **80.12** | 79.24 | 72.19 |

# G. Experimental Details

In addition to the setup described in Section 5.1, we provide comprehensive training details below. All experiments were conducted on a single RTX 4090 GPU and an Intel i9-13900KF CPU with 256 GB RAM. We adopt the SGD optimizer for all experiments, using a cosine learning rate decay schedule without gradient clipping.

For CIFAR-100, we train for 30 epochs with a batch size of 64, initial learning rate of 0.03, and weight decay of $1 \times 10^{-4}$ .

For ImageNet-R, we train for 20 epochs with a batch size of 32, initial learning rate of 0.05, and weight decay of $5 \times 10^{-4}$ .

For ImageNet-A, we train for 25 epochs with a batch size of 32, initial learning rate of 0.05, and weight decay of $5 \times 10^{-3}$ .

For DomainNet, we train for 5 epochs with a batch size of 64, initial learning rate of 0.03, and weight decay of $1 \times 10^{-4}$ .

