# OpenReview forum: "SABER: Continual Learning with Representation Conflict Management"
_ICML.cc/2026/Conference — ICML 2026 regular_

### Official Review · Reviewer_x7AS · 2026-03-08

**Soundness:** 3
**Presentation:** 3
**Significance:** 3
**Originality:** 3
**Overall Recommendation:** 4
**Confidence:** 4

**Summary:**

This paper proposes SABER (Subspace-Aligned Balanced Recomposition), a continual learning framework which explicitly manage representation conflicts rather than avoiding them. The method introduces three key components: subspace alignment, knowledge Recomposition, and knowledge balancing and reconstruction. SABER projects task-specific LoRA updates into a shared latent basis to ensure comparability across tasks, and then separates shared and task-specific representations into orthogonal components. They also proposed energy-aware balancing mechanism that adaptively integrates these components. Experiments on several continual learning benchmarks demonstrate that the approach achieves performance competitive with or superior to existing methods while maintaining strong parameter efficiency.

**Compliance With Llm Reviewing Policy:**

Affirmed.

**Final Justification:**

My concerns have been addressed. I keep my original positive score.

**Key Questions For Authors:**

Q1: The method implicitly assumes that different tasks share a common low-rank subspace. When tasks are highly dissimilar, is there any risk of heavy interference when reconstructing knowledge? Or can you provide any evidence that the proposed method still ourperforms others at different levels of task similairty?

Q2: In L249, can you provide the explicit implement of the statement "We set the hyperparameter α to define the proportion to preserve the task-common knowledge"? Also, in the discussion of "Abaltion study on $\alpha$", does the observation hold for other dataset and different task length(i.e., $T$)? If not, how to determine this hyperparameter?

**Limitations:**

yes

**Strengths And Weaknesses:**

Strengths:

1. This paper propose a new framework from the perspective of managing conflict rather than avoiding it. This is a novel perspective and potentially encourages further CL algorithm design.

2. This paper is organized and presented well. Authors proposed their method and explain the motivation. Their method SABER is practical and easy to implement.

3. The provided experiments are sufficient. The paper presents extensive experimental results that demonstrate the superior performance of the proposed method. Moreover, the authors provide analyses of the hyperparameter $\alpha$ as well as the computational complexity, which further strengthens the empirical evaluation.

Weakness:

1. This paper lacks discussion about task similairy. A detailed question refers to the question section.

---

> ### Author Rebuttal · Authors · 2026-03-29
>
> We thank Reviewer x7AS for the positive feedback and helpful suggestions.
>
> ### Weaknesses
>
> > **W3.1 & Q3.1**: Impact of task similarity, especially for highly dissimilar tasks.
>
> **AW3.1 & AQ3.1**: SABER does not rely on task similarity. The subspace alignment step maps all task-specific LoRA updates into a shared coordinate system using fixed orthogonal bases, which is independent of how similar the tasks are.
>
> During reconstruction, task-specific components are explicitly projected to be orthogonal to the shared subspace. This ensures that even when tasks are very different, their unique information is separated into non-overlapping directions, preventing destructive interference.
>
> Empirically, this behavior is consistent across datasets with different levels of similarity. CIFAR100 has more structured semantic overlap, while DomainNet is more diverse and fine-grained. SABER improves over prior methods in both cases, suggesting that the method remains stable regardless of task similarity.
>
> ---
> ### Questions
> > **Q3.2**: Implementation of α and its robustness across datasets and task lengths.
>
> **AQ3.2**: The hyperparameter α controls how much of the merged LoRA update is treated as shared (task-common) knowledge. We compute the SVD of $W_{\text{sum}} = \sum_{t=1}^T W_t$, and retain the top-$k$ components with $k = \alpha \cdot \mathrm{rank}(W_{\text{sum}})$.
> These top-(k) components form the shared subspace, while the remaining directions are assigned to task-specific components.
>
> We evaluate α across multiple datasets and task lengths (Table R3). The results show that performance is stable over a broad range of values. For datasets with higher similarity (e.g., CIFAR100), larger α performs slightly better, indicating stronger shared structure. For more diverse datasets (e.g., ImageNet-A, DomainNet), smaller α is preferable, as excessive sharing can introduce interference.
>
> Across all settings, α=0.6 provides a good balance between shared and task-specific knowledge and works consistently well.
>
> **Table R3: Accuracy of α across different datasets and task length**
> |**α**|**0.2**|**0.4**|**0.6**|**0.8**|**1.0**|
> |-|-|-|-|-|-|
> |CIFAR100|90.12|91.85|92.51|**92.63**|89.74|
> |ImageNet-R|81.69|84.12|**85.98**|85.50|76.89|
> |ImageNet-R(20steps)|79.24|82.37|**84.01**|83.85|74.12|
> |ImageNet-A|67.21|69.54|**70.06**|69.31|63.42|
> |DomainNet|76.83|79.67|**80.12**|79.24|72.19|

---

> > ### Author Rebuttal · Reviewer_x7AS · 2026-04-01
> >
> > The experimental results look solid, and the interpretation of $\alpha$ is intuitive and well-motivated. Overall, I believe this is a valuable contribution worth sharing with the CL community. I will maintain my original score.

---

> > > ### Author Response · Authors · 2026-04-02
> > >
> > > Thank you for your positive feedback and evaluation of our work. We sincerely appreciate your dedication and hard work throughout the review process. Wishing you a pleasant day ahead.

---

### Official Review · Reviewer_hhmR · 2026-03-13

**Soundness:** 3
**Presentation:** 3
**Significance:** 2
**Originality:** 3
**Overall Recommendation:** 4
**Confidence:** 4

**Summary:**

SABER proposes a post-training approach to merge per-task LoRA modules for class-incremental learning. The key insight is that naively summing LoRA updates amplifies shared directions while destroying task-specific ones. SABER addresses this by: (1) aligning all task LoRAs into a shared subspace via frozen orthogonal A matrices and SVD-derived output bases, (2) decomposing the sum into task-common (top-k SVD) and task-specific (residual SVD) components, (3) orthogonalizing and rebalancing their singular values before reconstruction. Evaluated on 4 CIL benchmarks with ViT-B/16, showing strong results over recent LoRA-based CL methods.

**Compliance With Llm Reviewing Policy:**

Affirmed.

**Final Justification:**

After reviewing the rebuttal and the other reviewers' feedback, I will keep my original positive score.

**Key Questions For Authors:**

- How does SABER perform when tasks are very similar (e.g., overlapping classes) vs. very diverse? Does $\alpha$ need to change?
- The SVD of Bstack is computed after all T tasks. How would SABER work in an online/streaming setting where T is not known in advance?
- Please see Weakness 1-2.

**Limitations:**

Yes

**Strengths And Weaknesses:**

**Strengths:**
- The subspace alignment idea is well-motivated and theoretically grounded.
-  The pipeline is conceptually clear: align, decompose, balance, reconstruct.
- Table 3 clearly shows each component's contribution and their synergy.

**Weaknesses:**
- The "conflict management" framing is overclaimed. The paper frames SABER as "managing representational conflict" rather than "avoiding interference." But what it actually does is: (a) enforce strict input-space orthogonality via frozen A (which is interference avoidance), and (b) post-hoc decompose+merge the output updates. The decomposition treats "conflict" as something to be separated and orthogonalized away, not something to be exploited. I don't see how this fundamentally differs from the avoidance paradigm — it's just avoidance at merge time rather than training time.
- The α hyperparameter controls the most critical part of the method. The α determines how much of the merged matrix is treated as "common" vs "specific" knowledge. This is essentially the entire algorithm's core decision, yet it's set to a fixed 0.6 across all datasets.
- The method doesn't scale well with T.  For ViT with d=768, r=10, you can have at most T = 76 tasks before the construction breaks.

---

> ### Author Rebuttal · Authors · 2026-03-29
>
> We thank Reviewer hhmR for the insightful feedback and for recognizing the motivation, design, and ablations of our method.
>
> ### Weaknesses
> > **W2.1**: The “conflict management” framing may be overclaimed.
>
> **AW2.1**: We agree that the current wording may overstate this point. Our goal is to highlight that SABER explicitly models and resolves interactions between task representations through structured alignment and decomposition, rather than relying solely on isolation. We will revise the wording to better reflect this distinction.
>
> ---
> > **W2.2 & Q2.1**: Role of α in balancing common vs. specific knowledge across tasks.
>
> **AW2.2 & AQ2.1**: We conducted a sensitivity analysis of α across all benchmarks (Table R2). The results show that SABER performs robustly over a wide range of values.
>
> For datasets with higher inter-task similarity (e.g., CIFAR100), larger α performs slightly better, indicating stronger reliance on shared components. For more diverse datasets (e.g., ImageNet-A, DomainNet), smaller α is preferred, as excessive sharing can introduce mild interference. Across all settings, α=0.6 provides a stable and consistently strong default.
>
>
> **Table R2: Accuracy of different datasets and α**
> |**α**|**0.2**|**0.4**|**0.6**|**0.8**|**1.0**|
> |-|-|-|-|-|-|
> |CIFAR100|90.12|91.85|92.51|**92.63**|89.74|
> |ImageNet-R|81.69|84.12|**85.98**|85.50|76.89|
> |ImageNet-R(20steps)|79.24|82.37|**84.01**|83.85|74.12|
> |ImageNet-A|67.21|69.54|**70.06**|69.31|63.42|
> |DomainNet|76.83|79.67|**80.12**|79.24|72.19|
>
> ---
>
> > **W2.3**: Scalability with respect to T (limited to 76 tasks for ViT-B/16).
>
> **AW2.3**: The current design limits T to 76 due to the fixed orthogonal basis. In practice, this is well beyond the scale of standard offline CIL benchmarks, where T is typically below 25 and rarely exceeds 50.
>
> If larger T is required, one can reinitialize a new orthogonal basis for later tasks. While this may introduce minor trade-offs, such scenarios are outside existing continual learning evaluation settings.
>
> ---
> ### Questions
> > **Q2.2**: Applicability to online/streaming settings where T is unknown.
>
> **AQ2.2**: SABER is designed for the offline setting. For online scenarios, a practical extension is to apply SABER periodically over a sliding window of recent tasks (e.g., every 5–10 tasks), performing alignment and recomposition within each window. This provides a feasible adaptation, though it introduces a trade-off between local adaptability and global consistency and is beyond our current scope.

---

> > ### Author Rebuttal · Reviewer_hhmR · 2026-04-02
> >
> > Thanks for the detailed rebuttal. The authors have addressed my concerns, and I will maintain my positive score.

---

> > > ### Author Response · Authors · 2026-04-03
> > >
> > > Thank you for your valuable suggestions and constructive comments on our manuscript. We sincerely appreciate your time and effort in providing insightful feedback. Wishing you a pleasant day ahead.

---

### Official Review · Reviewer_ni7H · 2026-03-13

**Soundness:** 3
**Presentation:** 3
**Significance:** 2
**Originality:** 2
**Overall Recommendation:** 3
**Confidence:** 3

**Summary:**

This paper studies class-incremental continual learning(CL) under the PEFT/LoRA setting. The main motivation is that, instead of simply avoiding interference across sequential tasks, continual adaptation should explicitly manage the representational conflicts that arise among task-specific updates. Therefore they proposes a structured LoRA merging framework that first trains task-specific LoRA branches for each task and then combines them through a three-stage post-processing pipeline. The method performs Subspace Alignment to project task-specific LoRA updates into a shared input-output subspace, Knowledge Recomposition to decompose the aligned updates into task-common and task-specific components, and Knowledge Balancing to rescale task-specific singular values(no individual task dominates the final merged model). Experiments show that SABER achieves competitive or superior performance compared with prior LoRA-based baselines while remaining parameter-efficient. Overall, the paper’s main contribution is to cast LoRA-based continual learning as a problem of structured representation conflict management and to introduce a corresponding merge-based solution.

**Compliance With Llm Reviewing Policy:**

Affirmed.

**Final Justification:**

Technically solid with reasonable clarifications, but limited evaluation scope and reliance on a task number assumption weaken its generality

**Key Questions For Authors:**

1. Requirement of knowing the task horizon in advance.
The Subspace Alignment step appears to allocate non-overlapping row blocks from a shared orthogonal basis Pfor each task via Eq. (5), and the aligned subspace is later defined using P_T. Does SABER therefore require the total number of tasks T, or at least an upper bound on T, to be known in advance? If so, this seems to make the method more suitable for a closed-world continual learning setting than for an open-ended online setting. It would be helpful if the authors could clarify this assumption explicitly and discuss what happens when the actual number of tasks exceeds the pre-allocated budget.

2. Storage of historical task-specific LoRA branches.
From Algorithm 1 and Eqs. (6)-(18), it seems that the method needs access to all historical task-specific LoRA parameters AtBt}t=1Tin order to compute the shared basis, perform alignment, and then conduct the final recomposition. Does SABER need to store all previous LoRA branches throughout training until the final merge stage? If yes, what is the total memory/storage overhead during training and deployment, and how does it compare to other multi-branch methods?

3. Generality beyond the current experimental scope.
The current experiments are mainly conducted on ViT-B/16 and on vision CIL benchmarks. Could the authors comment on how well the method is expected to generalize to broader settings, such as larger numbers of tasks, larger backbones, or continual fine-tuning scenarios for LLMs/VLMs?

4. Justification of the common/specific knowledge decomposition.
A central step of the method is the assumption that, after alignment, the dominant singular directions of the summed LoRA updates correspond to task-common knowledge, while the residual components correspond to task-specific knowledge. This is an interesting idea, but the paper currently provides limited evidence for why this decomposition should reliably hold in general. Could the authors provide more empirical or theoretical justification for this assumption？

**Limitations:**

Yes

**Strengths And Weaknesses:**

Strengths:
1. Clear and well-motivated problem framing. The paper identifies an important limitation of prior LoRA-based CL methods: mainly attempt to avoid interference rather than explicitly manage the representational conflicts. Thus, the forward transfer will be prevented.

2. Technically coherent method design. The framework is well structured, with the three components. In particular, the method is more than a simple heuristic merge rule, as it attempts to first align task-specific LoRA updates into a shared space and then separate/common-task-specific information before reconstruction.

3. Useful ablation studies. The paper includes ablations on the main components (SA, KR, KB) and also studies the effect of different initialization strategies for A_t. These analyses help support the claim that each component contributes meaningfully to the final performance.

Weaknesses:

1. The method appears to assume an offline/closed-world continual learning setting.
SABER seems to require knowing the total number of tasks or at least an upper bound, in advance, since the shared orthogonal basis P\ is used to allocate non-overlapping rows for different tasks, and the final recomposition step also relies on jointly processing all task-specific LoRA branches. This makes the method less suitable for open-ended or truly online continual learning.

2. Scalability and practicality are not fully clarified. Although the paper discusses efficiency, the method still relies on storing multiple task-specific LoRA branches and performing global post-hoc operations such as SVD-based alignment and recomposition. Therefore, the proposed method is more like solving the problem of merging pre-planned multi-task knowledge models than the problem of continuous learning.

3. The proposed method is based on LoRA and is independent of the model and specific task. However, the paper seems to only validate its performance on the visual CIL task, which limits the generalization of the method.

---

> ### Author Rebuttal · Authors · 2026-03-29
>
> We thank Reviewer ni7H for the careful and insightful review, and for recognizing our clear problem framing and technical design.
>
> ### Weaknesses
>
> > **W1.1 & Q1.1**: Requirement of knowing the total number of tasks (T) (or an upper bound) in advance.
>
> **AW1.1 & AQ1.1**: Thank you for highlighting this important point. Our current framework assumes that the total number of tasks (or a reasonable upper bound) is known in advance. We explicitly acknowledge this in the Conclusion, as SABER is designed for the **offline class-incremental learning** setting. This is consistent with the **standard offline CIL protocol** adopted by nearly all baselines we compare against (e.g., L2P [1]).
>
> If the actual number of tasks exceeds the pre-allocated budget, a practical extension is to periodically reinitialize the procedure with an expanded basis. While this falls outside our current scope, our contribution focuses on a principled framework for managing representational conflicts within the widely adopted offline setting.
>
> ---
> > **W1.2 & Q1.2**: Storage of historical task-specific LoRA branches and associated overhead.
>
> **AW1.2 & AQ1.2**: SABER follows the standard offline CIL protocol, where task-specific knowledge is retained without replay. This is consistent with prior methods: L2P [1] stores prompts, InfLoRA [2] stores gradient spaces, and CL-LoRA [3] stores LoRA branches.
>
> As shown in Table R1, SABER requires 4.63 MB during training, comparable to CL-LoRA (4.06 MB) and significantly lower than InfLoRA (24.33 MB). Importantly, SABER incurs **zero deployment overhead**, as all parameters are consolidated into the base model after recomposition.
>
> **Table R1: Storage overhead of training and deployment excluding the frozen backbone (MB)**
> |**Methods**|**Training**|**Deployment**|
> |-|-|-|
> |InfLoRA|24.33|0|
> |CL-LoRA|4.06|4.06|
> |**SABER**|**4.63**|**0**|
>
> ---
> > **W1.3 & Q1.3**: Generalization beyond vision CIL to larger backbones or LLMs/VLMs.
>
> **AW1.3 & AQ1.3**: Since SABER operates directly in the LoRA parameter space, it is backbone-agnostic and naturally extends to LLMs (e.g., aligning instruction-tuning LoRAs) and VLMs (e.g., merging vision-language adapters). A key challenge in these settings is increased task and modality heterogeneity, which may require more adaptive conflict modeling. We leave this as future work.
>
> ---
> ### Questions
> > **Q1.4**: Justification for decomposing aligned updates into task-common (top singular directions) and task-specific components.
>
> **AQ1.4**: In our formulation, task-specific knowledge is defined as the component of each LoRA that is orthogonal to the shared subspace, rather than simply the residual of summed updates.
>
> When summing aligned low-rank updates, directions consistently shared across tasks are reinforced, while task-specific variations tend to cancel or remain orthogonal. As a result, dominant singular directions capture the most consistent cross-task structure. This is supported by prior work [4] and by the Eckart–Young theorem [5], which guarantees that top singular components provide the optimal low-rank approximation. Empirically, isolating these components improves cross-task integration while preserving task-specific information, as reflected in our ablations.
>
> ---
> [1] Learning to prompt for continual learning (CVPR'22), Wang et al.
>
> [2] Inflora: Interference-free lowrank adaptation for continual learning (CVPR'24), Liang et al.
>
> [3] CL-LoRA: Continual low-rank adaptation for rehearsal-free class-incremental learning (CVPR'25), He et al.
>
> [4] Task singular vectors: Reducing task interference in model merging (CVPR'25), Gargiulo et al.
>
> [5] The approximation of one matrix by another of lower rank (1936), Eckart et al.

---

> > ### Author Rebuttal · Reviewer_ni7H · 2026-04-03
> >
> > After reading the rebuttal, I still view my main concerns as only partially resolved or unresolved, and I do not think they can be adequately addressed within a short rebuttal because they concern the core formulation and scope of the method.
> >
> > 1.	The rebuttal seems to conflate a finite benchmark protocol with a method-level requirement. In standard offline class-incremental learning benchmarks, the evaluation sequence is often pre-defined and finite, so the total number of tasks may be known to the experimenter. However, this does not imply that offline CIL methods must structurally depend on knowing the task horizon in advance. Many prior methods (L2P,DualPrompt,InfLoRA) can operate without explicitly allocating parameter blocks per task or requiring the total task count as part of the method definition. In contrast, SABER appears to rely on a pre-specified task budget (or upper bound) in its core formulation. Therefore, I view this as a substantive method-specific limitation rather than a generic property of offline CIL.
> >
> > 2.	My concern about the storage of historical LoRA branches was not really answered directly. The author only list the MB of storage, but do not explain the details and compare with other lora-based methods, I can not find any supporting evidence.
> >
> > 3.	Regarding generality, the authors mainly state that validation on larger backbones or LLM/VLM settings is future work. That is understandable, but it also means that the paper’s broader framing is still insufficiently supported by the current evidence.

---

> > > ### Author Response · Authors · 2026-04-03
> > >
> > > > **Q1**: The rebuttal seems to conflate a finite benchmark protocol with a method-level requirement. In standard offline class-incremental learning benchmarks, the evaluation sequence is often pre-defined and finite, so the total number of tasks may be known to the experimenter. However, this does not imply that offline CIL methods must structurally depend on knowing the task horizon in advance. Many prior methods (L2P,DualPrompt,InfLoRA) can operate without explicitly allocating parameter blocks per task or requiring the total task count as part of the method definition. In contrast, SABER appears to rely on a pre-specified task budget (or upper bound) in its core formulation. Therefore, I view this as a substantive method-specific limitation rather than a generic property of offline CIL.
> > >
> > > **AQ1**: We would like to respectfully clarify that the reviewer may misunderstand the problem setting of offline CIL. The T serves merely as a capacity upper bound to prevent resource exhaustion (similar to memory allocation), rather than a strict prerequisite for the algorithm to function. If T is required to have no upper bound at all, the situation are outside existing offline CIL evaluation settings, and all existing offline CIL methods (e.g., L2P, DualPrompt, InfLoRA) would encounter issues of poor performance and excessive storage overhead in this situation.
> > >
> > > When T exceeds a certain amount (i.e., 76), all methods can still operate. For SABER, we can initialize a new orthogonal matrix P for subsequent tasks, albeit facing a trade-off between performance and scalability as mentioned in AW2.3.
> > >
> > > When T falls within the certain amount, all methods can fully operate without knowing the actual number of T. When a new task arrives, we can simply allocate unused rows in the orthogonal matrix P for the new task.
> > >
> > > ---
> > > > **Q2**: My concern about the storage of historical LoRA branches was not really answered directly. The author only list the MB of storage, but do not explain the details and compare with other lora-based methods, I can not find any supporting evidence.
> > >
> > > **AQ2**: For our backbone ViT-B/16, we have feature dimension d=768, rank r=10, layer number L=12. For task number T=10, FP32(4 bytes per parameter), the calculation process of training storage overhead of the LoRA-based method (i.e., InfLoRA, CL-LoRA, and SABER) are shown in the Table R3.
> > >
> > > **Table R3: Storage overhead of training excluding the frozen backbone (MB)**
> > > |**Methods**|**Training**|**Training when T=10**|**Deployment**|
> > > |-|-|-|-|
> > > |InfLoRA|$Td^2+2drL$|$10\times 768^2+2\times 768\times 10 \times 12 \approx24.33MB$|0|
> > > |CL-LoRA|$2dr\times(L/2)+2dr\times(L/2)\times T$|$2\times 768\times10\times6+2\times768\times10\times6\times10\approx4.06MB$|4.06MB|
> > > |SABER|$d(d-1)/2+drLT$|$768\times(768-1)/2+768\times10\times12\times10\approx4.63MB$|0|
> > >
> > > Specifically, for InfLoRA, $Td^2$ represents the stored historical task gradient information, and $2drL$ denotes the individual branches that need to be extended for each task. For CL-LoRA, $2dr\times(L/2)$ denotes the shared module for all tasks, while $2dr\times(L/2)\times T$ indicates the individual module for each task. In SBAER, $d(d-1)/2$ stands for the minimum storage space required by the orthogonal matrix P, and $drLT$ represents the LoRA matrix that needs to be stored additionally for each task.
> > >
> > > After training, both InfLoRA and SABER can integrate the LoRA module into the backbone for deployment, whereas CL-LoRA must retain all modules.
> > >
> > > ---
> > > > **Q3**: Regarding generality, the authors mainly state that validation on larger backbones or LLM/VLM settings is future work. That is understandable, but it also means that the paper’s broader framing is still insufficiently supported by the current evidence.
> > >
> > > **AQ3**: We respectfully disagree with the premise that the empirical scope undermines the method's generality. SABER’s generalizability is theoretically grounded as a LoRA-based method. It is fundamentally architected as a LoRA-based extension which is highly compatible with the fine-tuning of LLM/VLMs, and its orthogonal subspace mechanism relies on scale-agnostic linear algebra principles. The evaluation of SABER is aligned with current CIL research that most methods (e.g., L2P, DualPrompt, InfLoRA, CL-LoRA) validate on mid-scale backbones, which further supports the reasonableness of our empirical scope.
> > >
> > > ---
> > > Given that the questions raised by the reviewer appear to stem from misunderstandings, we respectfully request that you carefully consider our clarifications. Additionally, we sincerely request that you reconsider your rating in light of the clarifications and evidence we have provided.

---

### Decision · Program_Chairs · 2026-04-30

**Decision:**

Accept (regular)

**Comment:**

Reviewers agreed that the paper is technically coherent, well motivated, and empirically solid within the evaluated LoRA-based class-incremental learning setting. In particular, they found the subspace alignment, decomposition, and balancing pipeline reasonably well supported by the ablations, and the rebuttal appears to have addressed the main concerns of the more positive reviewers, especially regarding the role of the balancing hyperparameter, task similarity, and the framing of “conflict management.”

The main remaining concerns are about scope rather than correctness. In particular, SABER currently assumes an offline setting with a preallocated capacity or task budget, requires retaining task-specific LoRA branches during training, and is evaluated only in visual class-incremental learning. These limitations do narrow the demonstrated generality of the method. However, the discussion suggests that these are better viewed as reasonable restrictions of the present submission than as fatal flaws in the core method.

Overall, I view this as a borderline but positive paper. The method is well structured, empirically competitive in its intended setting, and supported by a coherent technical story. While broader evaluation beyond the current offline visual CIL setup would strengthen the paper, I lean toward acceptance.